# Evaluation of Equisetin as an Anti-Microbial and Herbicidal Agent from Endophytic Fungus *Fusarium* sp. JDJR1

**Wei Wei** [1,†], **Ping Chen** [2,†], **Babar Khan** [3,†], **Kailin Tian** [2], **Yang Feng** [4], **Bei Lv** [1], **Nasir Ahmed Rajput** [5], **Wei Yan** [2], **Yonghao Ye** [2] and **Guiyou Liu** [1,*]

1   School of Life Sciences and Chemical Engineering, Jiangsu Second Normal University, Nanjing 211200, China; weiw@jssnu.edu.cn (W.W.); lvbei@jssnu.edu.cn (B.L.)
2   The Sanya Institute, Nanjing Agricultural University, Sanya 572000, China; 2021102128@stu.njau.edu.cn (P.C.); 2018102135@njau.edu.cn (K.T.); yanwei@njau.edu.cn (W.Y.); yeyh@njau.edu.cn (Y.Y.)
3   State Key Laboratory of Bioelectronics, School of Biological Science and Medical Engineering, Southeast University, Nanjing 210096, China; 101300304@seu.edu.cn
4   Faculty of Animal Science and Technology, Yunnan Agricultural University, Kunming 650201, China; 2022110092@stu.ynau.edu.cn
5   Department of Plant Pathology, University of Agriculture, Faisalabad 38000, Pakistan; nasir.ahmed@uaf.edu.pk
*   Correspondence: lgy@jssnu.edu.cn
†   These authors contributed equally to this work.

**Abstract:** Equisetin was isolated from endophytic fungus *Fusarium* sp. JDJR1 via bioassay-guided isolation, and it was evaluated as an anti-microbial and herbicidal agent. Bioactivity assessments revealed that equisetin exhibited a broad spectrum of fungicidal and anti-bacterial activity against various plant pathogens. The $EC_{50}$ values against *Botrytis cinerea*, *Fusarium graminearum*, *Sclerotinia sclerotiorum*, and *Rhizoctonia solani* ranged from 10.7 to 21.0 μg/mL. Notably, equisetin demonstrated activity against *Xanthomonas oryzae* pv. *oryzicola*, *Xanthomonas oryzae* pv. *oryzae* and *Pseudomonas solanacearum* with an MIC range of 4–16 μg/mL, surpassing the efficacy of the positive control: streptomycin sulfate. Equisetin, at a concentration of 100 μg/mL, could completely inhibit the spore germination of *F. graminearum*. In in vivo protection experiments, the protective efficacy of equisetin against *B. cinerea* on tomato fruits at 200 μg/mL reached 72.9%. Furthermore, in herbicidal activity tests that use the Petri dish bioassay method, equisetin had a good inhibitory effect on the root growth of weeds. At a concentration of 100 μg/mL, the inhibition rates for *Echinochloa crusgalli* and *Eclipta prostrata* root were 98.8 and 94.4%, respectively.

**Keywords:** equisetin; endophytic fungus; anti-microbial activity; herbicidal activity





## 1. Introduction

Plant diseases and weeds are major threats to the safety and stability of global agriculture production, and they can cause serious decreases in crop yield and quality [1]. Chemical control using synthetic pesticides is one of the most common methods for managing these pests. However, the widespread use and misuse of synthetic chemicals have caused potential hazards to humans, animals, and the environment. Meanwhile, the rapid development of pesticide resistance has led to the required development of pesticides with high efficiency, low toxicity, and eco-friendly attributes. Therefore, there is an urgent need to develop novel and effective anti-microbial and herbicidal agents to control agricultural diseases and weeds [2].

Nature has provided natural products as a rich source of diverse bioactive material, which has inspired the development of drugs/pesticides in medicinal and agricultural industry [3]. Plant endophytes are microorganisms residing inside the healthy plant tissues, and they possess no noticeable disease symptoms relative to host plants. They have been proven to be abundant resources of various natural products with different

bioactivities [4–6]. In addition, endophytes have emerged as potential producers of the bioactive constituents of plant origins. Taxol, one of the best-selling anticancer drugs originating from the plant *Taxus brevifolia*, was reported to be produced by its endophyte *Taxomyces andreanae* [7]. *Phoma glomerata* D14, an endophytic fungus from the traditional Chinese herb *Salvia miltiorrhiza*, is capable of generating salvianolic acid C, which is the same bioactive compound as the plant [8].

Our lab has carried out substantial research work on natural products from endophytes and made progress [9–11]. For example, the endophytic *Nigrospora chinensis* GGY-3 produced a tropolone compound stipitaldehyde, which exhibited broad-spectrum inhibition activity against pathogenic fungi and bacteria, especially against *Phytophthora capsici*, with an $EC_{50}$ value of 0.83 μg/mL and *Xanthomonas oryzae* pv. *oryzicola* (*Xoo*) and a minimum inhibitory concentration (MIC) value of 4.0 μg/mL [9]. Three new eremophilane sesquiterpenes and two new benzene derivatives were isolated from endophytic fungus *Microdiplodia* sp. WGHS5 with potent cytotoxic activities [10]. Five novel polyketide derivatives, isotalaroflavone, (+/−)-50-dehydroxytalaroflavone, (+)-talaroflavone, and bialternacin G, along with five known compounds, were obtained from the *Cercis chinensis*-derived fungus *A. alternata* ZHJG5. Some of these compounds showed pronounced antibacterial activity against phytopathogenic bacteria, with MIC values ranging from 0.5 to 64 μg/mL [11].

Recently, we screened a collection of endophytes isolated from plants collected in Laoshan National Forest Park, Nanjing City, and revealed that *Fusarium* sp. JDJR1 displayed potent antifungal activity. The *Fusarium* species plays an important role in the environment and is widely distributed in nature; it is usually found in soil, plant parts, plant debris, and other organic substrates from various regions. The widespread distribution of *Fusarium* species may result from the ability of these fungi to grow on a wide range of substrates and their efficient mechanisms for dispersal. Common *Fusarium* species include *F. equiseti*, *F. lateritium*, *F. graminearum*, *F. moniliform*, *F. nivale*, *F. oxysporum*, *F. solani*, etc. Many have been reported as plant pathogens, nonpathogenic endophytes, or saprobes, which may be explained, in part, by their capacity to colonize diverse ecological niches in most geographic areas [12].

As plant pathogenic fungi, *Fusarium graminearum*, for example, can infect wheat and cause wheat head blight, resulting in damage to wheat quality and yield. Its secondary metabolites, known as mycotoxins, pose a great threat to human and animal health [13]. Another example is *F. solani*, which can invade the interior of potatoes from wounds, causing an increase in the proportion of water loss, surface wrinkling, and the appearance of brown lesions and leading to the dry rot of potatoes. In the later stage, the interior of potatoes begins to rot, discolor, and form cavities filled with mycelium, greatly reducing the quality and safety of potatoes [14]. The phytopathogenic fungus *F. oxysporum* can infect crops such as cotton, tomatoes, and soybeans, giving rise to plant wilt disease. This pathogen can invade plants from the roots, resulting in the blockage of vascular bundles. Due to the inability of water and nutrients to be transmitted normally, plants will wither and die, causing substantial damage to economic crops [15].

In addition, *Fusarium* is also a common endophytic fungus that can synthesize novel secondary metabolites in the process of interacting with its host plant, helping the host plant avoid external invasion. Currently, research on metabolites from endophytic *Fusarium* fungi shows that their metabolites consist of different structural types, including ketones, terpenes, alkaloids, cyclic peptides, naphthoquinones, etc. These metabolites exhibit an array of bioactivities: for instance, antibacterial, insecticidal, anti-tumor, antiviral, and antioxidant activities [16–20].

The fermentation and extraction of our endophytic *Fusarium* sp. JDJR1 led to the discovery of a bioactive substance identified as equisetin. Equisetin was first isolated from white mold *Fusarium hererosporum* by Hesseltine and co-workers in 1974 and reisolated from *Fusarium equiseti* in 1979, which shows potent antibiotic activity [21,22]. It was reported that equisetin also exhibits HIV inhibitory activity, cytotoxicity, and anti-obesity activity [23]. The tetramic acid (2,4-pyrrolidinedione) ring system is a key structural

unit in equisetin [24,25]. The chemical synthesis and biological studies of equisetin have attracted considerable attention from synthetic chemists and biologists [26]. Nevertheless, the agrochemical applications of equisetin have been less reported. In this paper, the isolation, antimicrobial bioassay, and herbicidal assay of equisetin are described in detail.

## 2. Materials and Methods

### 2.1. Plant Material

Healthy parts of *Mirabilis jalapa* were collected from the Laoshan National Forest Park, Nanjing City, Jiangsu Province, China, in September 2018. The identity of the plant was confirmed via the voucher specimen number used by the park.

### 2.2. Isolation of Endophytic Fungal Strains from M. jalapa

The collected samples were carefully transported to the laboratory in sealed polythene bags to preserve their integrity for the subsequent isolation of endophytic fungi. In the laboratory, healthy plant parts, comprising leaves and stems, were cut into 1–2 cm segments. To ensure the elimination of external contaminants, the plant segments underwent a stringent surface sterilization process. This involved sequential washing with 75% ethanol (1 min), sterile distilled water (1 min), 75% ethanol (2 min), and sterile distilled water (2–3 min). Following sterilization, the plant parts were dried using sterile filter paper and then positioned on a potato dextrose agar (PDA) medium in Petri dishes. The prepared dishes were subsequently incubated for several days at a temperature of 25 °C. Colonies that emerged from the edges of the plant segments were meticulously sub-cultured on PDA plates via a series of transfers until pure cultures were obtained. A total of 5 fungal isolates were successfully purified via this process.

### 2.3. Fermentation, Extraction, and Antimicrobial Bioassay Screening

Each fungal isolate was cultured on four PDA-containing Petri plates (120 × 20 mm) for 6–8 days. When the fungi were close to sporulation, each fungus together with the medium was cut into 5 mm small plugs, and 10 plugs of each isolate were placed into 1000 mL Erlenmeyer flasks containing a 400 mL medium of potato dextrose broth (PDB), which were shaken at 120 rpm for 12–14 days at 28 °C. Fermentation broths were filtered, extracted with ethyl acetate (EtOAc) three times (broth/EtOAc, 1/1), and dried for crude extracts. The fungal crude extracts dissolved in dimethyl sulfoxide (DMSO) were tested using the mycelial growth inhibition method [27] (details were described in Section 2.7) against pathogenic fungal strains at 50 μg/mL. Strains with inhibition rates of more than 30% will be selected for further study. Boscalid was used as a positive control, and DMSO served as the negative control. Each test was repeated three times.

### 2.4. Fungal Strain

Fungal strain JDJR1 was selected for study to isolate and characterize the active compounds. Strain identification was carried out by observing the morphological character on the PDA medium and blasting its ITS sequence (GenBank Accession No. MW369578.1) in the NCBI database [10]. The fungus was deposited in the culture collection bank of the Laboratory of Natural Products and Pesticide Chemistry, Nanjing Agricultural University (NAU), Nanjing, Jiangsu, China.

### 2.5. Isolation of Equisetin from Fusarium sp. JDJR1 via Bioassay-Guided Fractionation

The crude EtOAc extract of the fungal strain JDJR1 was subjected to bioassay-guided chromatographic fractionations with reference to antifungal activity. Using this method, the fractions obtained from column chromatography were subjected to antifungal bioassays to detect the active fractions. The crude EtOAc extract (12 g) was subjected to silica gel chromatography (6 × 40 cm) using a gradient solvent system of $CH_2Cl_2$-MeOH (100:0, 100:1, 100:2, 100:4, 100:8, 100:16, and 0:100, flow rate = 40 mL/min) to yield five fractions (Fr1–Fr5), of which Fr3 exhibited antifungal activity (more than 50% inhibition rates at

50 µg/mL). The active fraction Fr3 was further fractionated via another silica column (petroleum ether/EtOAc, *v/v*, 30:70) to obtain fraction Fr3-3 (178 mg), which was finally purified using Sephadex LH-20 CC (MeOH) and semipreparative HPLC (MeOH/H$_2$O, *v/v*, 45:55, to obtain equisetin (20.5 mg) as a colorless oil.

### 2.6. Structural Elucidation of Equisetin

The structural determination of the isolated compound equisetin was carried out using mass and NMR spectral data. High-resolution electrospray ionization mass spectrometry (HR-ESI-MS) data were collected using an Agilent 6210 TOF LC−MS spectrometer (Agilent Technologies, Santa Clara, CA, USA). All nuclear magnetic resonance (NMR) experiments were recorded using the CDCl$_3$ solution on a Bruker Avance III 600 MHz NMR spectrometer (Bruker, Rheinstetten, Germany). Thin-layer chromatography (TLC) was performed on pre-coated silica gel GF$_{254}$ plates (Qingdao Haiyang Chemical Factory, Qingdao, China). Column chromatography (CC) was performed using silica gel (SiO$_2$, 100−200 and 200−300 mesh, Qingdao Marine Chemical Inc., Qingdao, China) and Sephadex LH-20 (Pharmacia Biotech, Uppsala, Sweden). Analytical-grade solvents were used in this study.

### 2.7. In Vitro Antifungal and Antibacterial Assays of Equisetin

The isolated compound was tested for antifungal and antibacterial activities against plant pathogenic fungi (*Botrytis cinerea*, *Fusarium graminearum*, *Sclerotinia sclerotiorum*, and *Rhizoctonia solani*) and phytopathogenic bacteria (*Xanthomonas oryzae* pv. *oryzicola*, *X. oryzae* pv. *oryzae*, *Staphylococcus aureus*, *Ralstonia solanacearum*, and *Bacillus subtilis*). All strains were provided by the Laboratory of Natural Products and Pesticide Chemistry, Nanjing Agricultural University.

The antifungal assay was conducted via the mycelial growth inhibition method. Pathogenic fungi were taken out from the preservation tube and revived 2–3 times on a PDA plate for later use. Compounds were dissolved in DMSO to prepare the stock solution before mixing with molten PDA below 60 °C. The initial concentration used for preliminary screening was set at 50 µg/mL. An equal amount of DMSO was tested as the blank control, and Boscalid served as the positive control. A fungal block (5 mm) was picked up from the edge of the pathogen colony and inoculated onto a PDA plate containing tested compounds. The plates were incubated at 25 °C in the dark. When the mycelium of the blank control grew to 5–6 cm, the colony diameter was measured using the crossover method. The percentage inhibition (%) was calculated as $(B − A)/(B − 5) × 100\%$, where A is the mycelial diameter (mm) in plates with compounds, B is the diameter (mm) of the negative control, and 5 is the diameter (mm) of fungal blocks made of a cork borer. The inhibition rate of the potent compounds was further evaluated at different concentration gradients, and the corresponding median effective concentration (EC$_{50}$) values were calculated according to the concentration-dependent curve.

The MTT method was used for the antibacterial activity test. The strains for testing were taken out from the −80 °C refrigerator and placed into an ice box. After thawing, strains were streaked on the NA plate and cultured at 28 °C for 2–3 days. Then, a single colony was picked using a sterilized toothpick and added to the NB liquid media, which was shaken at 28 °C and 150 rpm until the concentration of solution reached 10$^5$ CFU/mL. On a 96-well microplate, each well was added with 100 µL of NB medium, 98 µL of diluted bacterial solution, and 2 µL of tested compounds. Three replicates were performed, with streptomycin sulfate used as the positive control and 1% DMSO in the medium as the negative control. After incubating at 28 °C for 18 h, 10 µL thiazole blue (MTT) was added to the wells, and a color change was observed after 2–3 h. If the bacterial solution changes color, this indicates that bacterial growth cannot be inhibited at this concentration. Otherwise, it indicates that the compound has an inhibitory effect on bacteria. The MIC value was the lowest concentration of the compounds for which no blue color was seen.

### 2.8. Inhibitory Test against Spore Germination of F. graminearum

*F. graminearum* was inoculated into 3% mung bean soup medium and incubated at 25 °C and 120 rpm in the dark for about 7–10 days. The culture broth was filtered with three layers of lens paper, transferred into a 1.5 mL centrifuge tube, and centrifuged at 5000 rpm for 10 min to remove the supernatant. The filtrate was collected and mixed with sterile water, which was centrifuged 2–3 times before collecting spores. The mixed spore suspension was diluted to a concentration of $1 \times 10^5$ CFU/mL, as observed via a microscope, using a blood cell counting plate. In the sterilized centrifuge tube, a 40 μL spore suspension and 40 μL compound solution (200, 20, and 2 μg/mL) were added, mixed, and placed in a 25 °C incubator for about 6 h of dark with moisture. An aqueous solution containing 0.25% DMSO was used as the blank control. When the germination rate of the control group reached 85%, the germination of the treatment group was observed and recorded. The criterion for spore germination is as follows: The length of the spore tube is greater than half the length of the spore. In each group, at least 100 spores were observed under an optical microscope, and the number of germinated spores was recorded. The calculation method is as follows: germination inhibition rate = (spore germination rate of control group − spore germination rate of treatment group)/spore germination rate of control group × 100% [28].

### 2.9. Protective Efficacy of Test Compound against B. cinerea

Healthy tomato fruits with uniform color and size were selected for this experiment. Prepared aqueous solutions (0.05% Tween 80 and 0.5% DMSO) containing test compound concentrations of 100 and 200 μg/mL were spread evenly on the surface of tomato fruit with a small brush. The treatment without compounds served as the negative control, and Boscalid was used as the positive control. Each treatment was repeated three times and cultivated at 25 °C for 24 h. A small hole was pricked in the tomato fruit with a sterile needle, and the activated pathogenic fungal cake was inoculated into the pricked hole. After cultivating in the dark at 25 °C for 72 h, the diameter of the lesion was measured and the inhibition rate was calculated using the same formula in [29].

### 2.10. Wheat Head Blight Protection Assay

The wheat plants required for this experiment were sown in advance. Aqueous solutions (0.05% Tween 80 and 0.5% DMSO) containing test compounds were prepared with concentrations of 100 and 200 μg/mL, respectively. The blank control group used an aqueous solution without compounds, and fungicide tebuconazole served as the positive control at the same concentrations. Tested solutions were spread on the ears of wheat with the help of a small sprayer bottle. After 24 h, 10 μL of the spore suspension ($10^5$ conidia/mL) of *F. graminearum* was dropped into each wheat ear, and they were covered and kept moist for 48 h. After one week, the disease severity index (SI) and control effect were calculated using the formulas reported in [30].

### 2.11. Herbicidal Efficacy of Equisetin

*Echinochloa crusgalli* and *Euphorbia prostrata* seeds were selected for herbicidal assays using the Petri dish bioassay method. Weed seeds were soaked in a 5% NaClO solution for 10 min and then rinsed 2–3 times with sterile water. Compounds were prepared in aqueous solution concentrations of 200, 100, 50, and 25 μg/mL (containing 0.25% DMSO). Then, the 0.25% DMSO aqueous solution was tested as the blank control, and penoxsulam was used as the positive control. Two layers of filter paper were placed onto the Petri dish, and 2 mL of test solutions was added. Twenty weed seeds were placed into every dish, which was sealed with a polyethylene wrapping film with some small holes and placed in a growth chamber at 28 °C under 14 h of daylight and 10 h of darkness. Each treatment was repeated three times. After 7 and 10 days of cultivation, the root and shoot lengths of weed seeds were measured. The inhibition rates are calculated according to the following

formula: inhibition rate (%) = ((the length of control − the length of treatment)/the length of control) × 100 [31].

## 3. Results and Discussion

### 3.1. Screening of Fusarium sp. JDJR1

The antifungal activities of the crude extracts of the isolated endophytic fungi are given in Table 1. The results showed that the ethyl acetate extracts had different inhibitory effects on plant pathogenic fungi. Out of the five fungal extracts, strain JDJR1 was active against all pathogenic fungal strains (*R. solani* (50.9%), *F. graminearum* (43.3%), *S. sclerotiorum* (78.6%), and *P. capsici* (33.4%)). As a result, strain JDJR1 was selected as the target strain for in-depth chemical investigation.

**Table 1.** Antifungal activity of the crude extracts of endophytic fungal strains (%).

| Strains | *R. solani* | *F. graminearum* | *S. sclerotiorum* | *P. capsici* |
|---|---|---|---|---|
| JDJR1 | 50.9 | 43.3 | 78.6 | 33.4 |
| DWFCY1 | - | 20.6 | 91.4 | 27.3 |
| CCY3 | 50.9 | 30.0 | 31.1 | 33.0 |
| YTY1 | - | 15.8 | - | 7.7 |
| ZMLY1 | 7.7 | 24.7 | - | 4.1 |

Notes: "-" means no activity was observed.

### 3.2. Structure Elucidation of the Bioactive Metabolite from Fusarium sp. JDJR1

Equisetin was isolated as a colorless oil. On the basis of its HRESIMS data (found [M + H]$^+$, *m/z* 374.2335; calculated for $C_{22}H_{32}NO_4^+$, 374.2331), the compound was found to have $C_{22}H_{31}NO_4$ as its molecular formula, indicating eight degrees of unsaturation. The analysis of $^1$H and $^{13}$C NMR spectral data in CDCl$_3$ revealed four methyls ($\delta_C$ 14.0, 18.0, 22.5, and 27.2), four methylenes ($\delta_C$ 28.3, 35.7, 42.3, and 60.1), three double bonds ($\delta_C$ 100.3, 126.6, 127.2, 130.0, 130.8, and 190.8), and two ketones ($\delta_C$ 177.0 and 199.0). This suggested that the structure of the active compound was identical to the known equisetin (Figure 1) [32]. A comparison of $^{13}$C NMR values obtained in the present study for equisetin with already-reported data is given in Table 2.

**Table 2.** Comparison of $^{13}$C NMR data of equisetin from the present study (in CDCl$_3$) with published data.

| Position | Present Values | Reported Values |
|---|---|---|
| 1 | 190.8, C | 190.7, C |
| 2 | 48.8, C | 48.7, C |
| 3 | 44.9, CH | 44.9, CH |
| 4 | 126.6, CH | 126.6, CH |
| 5 | 130.0, CH | 129.9, CH |
| 6 | 38.5, CH | 38.5, CH |
| 7 | 42.3, CH$_2$ | 42.2, CH$_2$ |
| 8 | 33.5, CH | 33.5, CH |
| 9 | 35.7, CH$_2$ | 35.6, CH$_2$ |
| 10 | 28.3, CH$_2$ | 28.3, CH$_2$ |
| 11 | 39.9, CH | 39.9, CH |
| 12 | 14.0, CH$_3$ | 14.1, CH$_3$ |
| 13 | 127.2, CH | 127.1, CH |
| 14 | 130.8, CH | 130.8, CH |
| 15 | 18.0, CH$_3$ | 17.9, CH$_3$ |
| 16 | 22.5, CH$_3$ | 22.5, CH$_3$ |
| 17 | 177.0, C | 177.0, C |
| 18 | 100.3, C | 100.9, C |
| 19 | 199.0, C | 198.9, C |
| 20 | 66.5, CH | 66.3, CH |
| 21 | 60.1, CH$_2$ | 60.2, CH$_2$ |
| 22 | 27.2, CH$_3$ | 27.2, CH$_3$ |

**Figure 1.** Chemical structure of equisetin.

### 3.3. Antifungal and Bactericidal Potency of Equisetin

The inhibitory effect of equisetin on phytopathogenic fungi showed that the $EC_{50}$ values of equisetin against *B. cinerea*, *F. graminearum*, *S. sclerotiorum*, and *R. solani* were 10.7, 12.9, 17.1, and 21.0 μg/mL, respectively (Table 3). Equisetin was also screened for antibacterial activity against different bacteria. The results showed that equisetin had a wide antibacterial spectrum and could inhibit *Xanthomonas oryzae* pv. *oryzicola*, *X. oryzae* pv. *oryzae*, *S. aureus*, *R. solanacearum*, and *B. subtilis*, among which the inhibitory activities with respect to *Xanthomonas oryzae* pv. *oryzicola* and *B. subtilis* were the best, with the same MIC value of 4 μg/mL, which was better than the positive control streptomycin sulfate (Table 4).

**Table 3.** $EC_{50}$ of test compound against four plant-pathogenic fungi (μg/mL) [a].

| Compound | *B. cinerea* | *F. graminearum* | *S. sclerotiorum* | *R. solani* |
|---|---|---|---|---|
| Equisetin | 10.7 ± 2.1 | 12.9 ± 2.5 | 17.1 ± 6.7 | 21.0 ± 5.2 |
| Boscalid | 0.8 ± 0.2 | 6.9 ± 0.4 | 4.2 ± 0.3 | 0.4 ± 0.1 |

[a]: Data were presented as mean ± SD of three independent experiments.

**Table 4.** MIC of test compound against five bacteria (μg/mL).

| Compound | *Xoc* | *Xoo* | *S. a* | *R. s* | *B. s* |
|---|---|---|---|---|---|
| Equisetin | 4 | 16 | 16 | 16 | 4 |
| Streptomycin sulfate | 8 | 32 | 2 | 16 | 8 |

Note: *Xoc*: *Xanthomonas oryzae* pv. *oryzicola*; *Xoo*: *X. oryzae* pv. *oryzae*; *S. a*: *S. aureus*; *R. s*: *R. solanacearum*; *B. s*: *B. subtilis*.

### 3.4. Germination Inhibition of F. graminearum Spores

The results of the spore germination of *F. graminearum* showed that after 8 *h* of treatment, the spore germination rate of the blank control (water) and the negative control (0.25% DMSO) reached 98% and 96%, respectively, indicating that the addition of 0.25% DMSO had no effect on the germination of spores. At a concentration of 100 μg/mL, equisetin could completely inhibit spore germination, with a similar effect as Boscalid at 10 μg/mL. When tested at 10 and 1 μg/mL, the inhibition rates of equisetin reached 90.1% and 37.5%, respectively (Figure 2).

### 3.5. Protective Efficacy of Equisetin against Tomato Gray Mold and Wheat Head Blight

The potential of equisetin to control gray mold was tested on tomato fruits at concentrations of 100 and 200 μg/mL. The results showed that the compound had a protective efficacy of 47.7% and 72.9% on tomato fruit botrytis at 100 and 200 μg/mL, respectively. This suggests that equisetin significantly reduced the lesions on tomato fruits caused by *B. cinerea* (Table 5 and Figure 3).

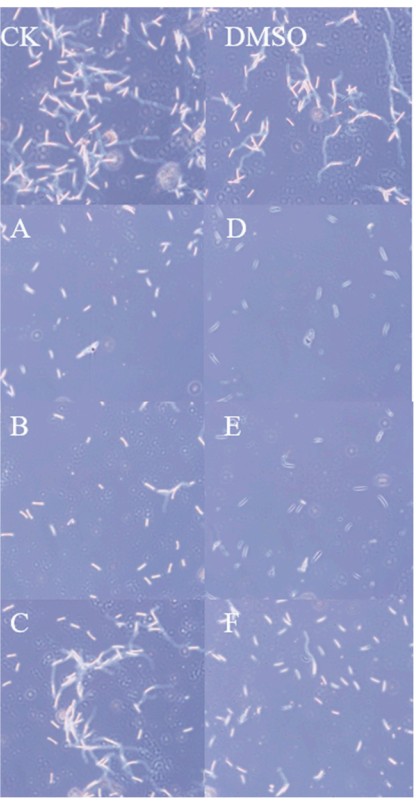

**Figure 2.** Effects of equisetin and Boscalid on *F. graminearum* spore germination at different concentrations. (**A–C**) are the microscope fields of *F. graminearum* spore germination treated with equisetin at concentrations of 100, 10, and 1 μg/mL, respectively; (**D–F**) are the microscope fields of *F. graminearum* spore germination treated with Boscalid at concentrations of 100, 10, and 1 μg/mL, respectively.

**Table 5.** Inhibition rate of compound equisetin against *B. cinerea* on tomato fruits (%) [a].

| Treatments | Concentration (μg/mL) | Inhibition Rate (%) |
|---|---|---|
| Equisetin | 100 | 47.7 ± 6.4 |
|  | 200 | 72.9 ± 3.1 |
| Boscalid | 100 | 71.0 ± 1.9 |
|  | 200 | 84.1 ± 1.8 |

[a]: Data were presented as mean ± SD of three independent experiments.

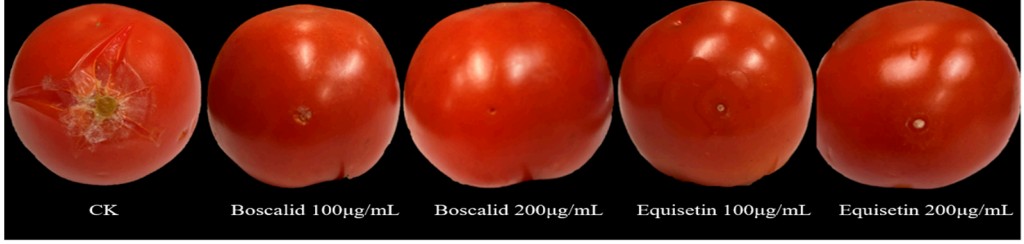

**Figure 3.** Protective effects of equisetin against *B. cinerea* on tomato fruits.

Equisetin was also selected for protective efficacy against wheat head blight. The test compound and positive tebuconazole control were used at the same concentrations of 200 and 100 μg/mL. The results showed that equisetin had a certain control effect on wheat head blight (51.5% at 200 μg/mL) (Table 6 and Figure 4).

**Table 6.** Protective efficacy of equisetin against wheat head blight.

| Treatments | Concentration (μg/mL) | Disease Index (%) | Control Effects (%) |
|---|---|---|---|
| Equisetin | 100 | 73 | 24.7 |
| | 200 | 47 | 51.5 |
| Tebuconazole | 100 | 38 | 60.8 |
| | 200 | 5 | 94.8 |
| CK | - | 97 | - |

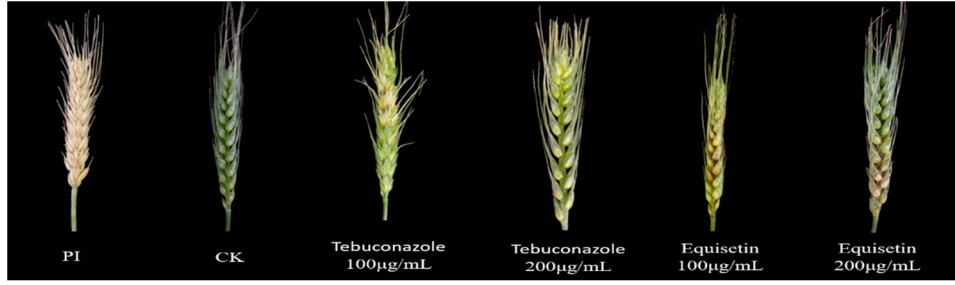

**Figure 4.** Protective effects of equisetin against wheat head blight.

### 3.6. Herbicidal Evaluation of Equisetin

The herbicidal activity against *Echinochloa crusgalli* and *Euphorbia prostrata* of the test compound was determined, and the results revealed that equisetin had obvious inhibitory effects on the roots of *E. prostrate* at various concentrations (200, 100, 50, and 25 μg/mL). The inhibition rate was observed on roots (98.8%) and shoots (41.6%) at 100 μg/mL, respectively (Table 7 and Figure 5). Equisetin also had a good inhibitory effect on the roots of *E. crusgalli*, and the inhibition rate was 98.1% on roots and 37.5% on shoots at a concentration of 200 μg/mL (Table 7 and Figure 6).

**Table 7.** Inhibition (%) of equisetin on the root length and shoot length of *E. crusgalli* and *E. prostrate* [a].

| Compound | Concentration (μg/mL) | *E. crusgalli* | | *E. prostrate* | |
|---|---|---|---|---|---|
| | | Shoot Inhibition (%) | Root Inhibition (%) | Shoot Inhibition (%) | Root Inhibition (%) |
| Equisetin | 200 | 37.5 ± 4.0 | 98.1 ± 0.5 | 70.9 ± 3.8 | 100 ± 0 |
| | 100 | 22.0 ± 1.8 | 94.4 ± 1.5 | 41.6 ± 5.7 | 98.8 ± 0.5 |
| | 50 | 27.2 ± 1.7 | 66.5 ± 2.1 | 26.1 ± 4.2 | 75.5 ± 0.8 |
| | 25 | 23.8 ± 5.0 | 54.5 ± 4.2 | 19.8 ± 0.9 | 21.0 ± 1.7 |
| Penoxsulam | 200 | 35.5 ± 3.3 | 96.8 ± 1.3 | 90.1 ± 1.6 | 85.1 ± 2.3 |
| | 100 | 36.9 ± 3.3 | 90.9 ± 1.2 | 80.4 ± 3.0 | 82.5 ± 2.1 |
| | 50 | 33.6 ± 4.3 | 87.8 ± 1.8 | 74.7 ± 2.8 | 78.7 ± 2.9 |
| | 25 | 30.5 ± 5.8 | 84.2 ± 2.8 | 72.4 ± 5.8 | 70.9 ± 2.5 |

[a]: Data were presented as the mean ± SD of three independent experiments.

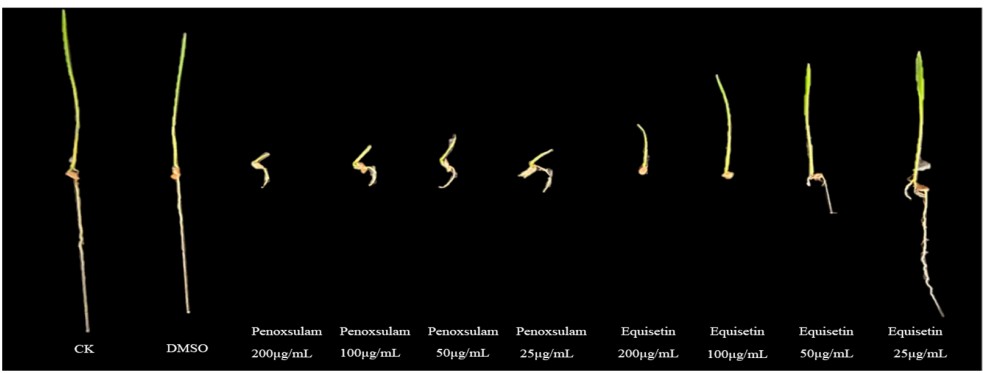

**Figure 5.** Inhibition (%) of the test compound on roots and shoots of *E. prostrate*.

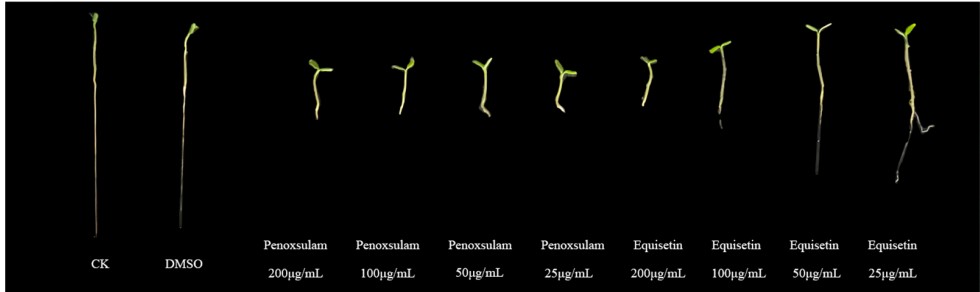

**Figure 6.** Inhibitory effect (%) of the test compound on the roots and shoots of *E. crusgalli*.

## 4. Conclusions

Plant diseases and weeds pose significant threats to agricultural production and product quality not only in China but also globally. The conventional approach to combating pests involves the use of pesticides, but the indiscriminate application of these chemicals has led to problems, such as increasing resistance in target organisms, the disruption of the ecosystem's balance, and environmental pollution. Consequently, there is an ongoing imperative to develop environmentally friendly pesticides that are characterized by high efficacy, low toxicity, and low residue. Natural products, due to their unique structures, various biological activities, and eco-friendly characteristics, play a pivotal role in the development of innovative pesticides. Plant endophytes, widely distributed in nature, generate metabolites with novel chemical structures and distinct biological activities during their interaction with the environment. These compounds, in turn, serve as valuable sources of lead pesticide compounds.

The development of natural products into pesticides can take various routes, including utilizing unmodified compounds, semi-synthesis, and synthetic mimics. The current study highlights equisetin as a promising antimicrobial and herbicidal compound with potential applications in crop protection. However, further research is imperative to elucidate its mode of action, optimize fermentation conditions to improve yield, and address challenges related to rapid separation and purification. In the realm of chemical synthesis, this compound is complex and contains many stereocenters, which increases synthetic costs and limits its practical application. Resolving this issue involves identifying the pharmacophore and simplifying the compound's structure. While natural products are generally considered safer, with low toxicity relative to non-target organisms and human health, a thorough examination of equisetin's toxicity is necessary before advancing into subsequent stages. Additionally, exploring the ecotoxicology of equisetin, although less concerning than synthetic compounds, remains crucial for comprehensive understanding, particularly if contemplating its use as a commercial pesticide.

**Author Contributions:** Conceptualization, W.W., P.C., B.K., K.T., W.Y., Y.Y. and G.L.; investigation, W.W., P.C., B.K., K.T., Y.F. and B.L.; writing—original draft preparation, W.W., P.C., B.K., N.A.R. and W.Y.; supervision, W.Y., Y.Y. and G.L.; project administration, W.Y., Y.Y. and G.L.; funding acquisition, W.Y., Y.Y. and G.L. All authors have read and agreed to the published version of the manuscript.

**Funding:** This research was funded by the Key Project of Natural Science Foundation of the Higher Education Institution of Jiangsu Province (19KJA430013), the Guidance Foundation, the Sanya Institute of Nanjing Agricultural University (NAUSY-ZD04), and the Natural Science Foundation of Jiangsu Province (BK20211214).

**Data Availability Statement:** Data are contained within the article.

**Conflicts of Interest:** The authors declare no conflict of interest.

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
