# Peer review of "Evaluation of Equisetin as an Anti-Microbial and Herbicidal Agent from Endophytic Fungus Fusarium sp. JDJR1"

_agronomy, doi:10.3390/agronomy14010031_

Round 1
Reviewer 1 Report
Comments and Suggestions for Authors
Major Comments:
- Abstract:
- The abstract provides a concise overview of the study. However, it would be beneficial to include the methodology used for herbicidal activity testing in the abstract as this is a significant aspect of the study.
- The EC50 and MIC values provided lack units in the abstract. Please ensure to include the appropriate units (e.g., μg/mL) for clarity.
- Introduction:
- The introduction provides a clear rationale for the study and highlights the importance of developing environmentally friendly pesticides. However, there is a need to better connect the introductory paragraphs to the specific focus of the current research on equisetin. It is suggested to provide a more direct transition from the general need for novel pesticides to the specific investigation of equisetin.
- Materials and Methods:
- The isolation of endophytic fungi from Mirabilis jalapa is well-described. However, the selection criteria for choosing Fusarium sp. JDJR1 among the isolated strains for further study should be provided.
- In Section 2.3, the method for mycelial growth inhibition is referenced but not described. Please provide details of this method for clarity.
- In Section 2.5, the bioassay-guided fractionation process is outlined, but additional details regarding the specific chromatographic conditions (e.g., column dimensions, flow rates) and the criteria used to determine active fractions would enhance reproducibility.
- Results:
- The results section is well-organized and presents key findings effectively. However, more discussion on the structural elucidation of equisetin would enhance the scientific content. Consider providing key NMR and mass spectrometry data for a comprehensive understanding.
- Discussion: Authors are suggested to revise the discussion section in separate heading. Results must be discussed logically and compared with literature as well.
- Conclusion:
- The conclusion effectively summarizes the findings but lacks specific recommendations for future research or potential applications of equisetin in practical agriculture.
- Limitations of the study: Authors should add limitations of the study as well
- General comments:
- Ensure consistency in reporting units throughout the manuscript. For instance, some concentrations are reported in μg/mL, while others are in %. Furthermore, check the clarity of the manuscript especially in sections related to methodology and results. Ensure that the experimental procedures are described in sufficient detail to allow for reproducibility.
Spelling and grammatical mistakes should be rechecked throughout the manuscript.
Reviewer 2 Report
Comments and Suggestions for Authors
Abstract:
- You may consider lightening up this section by reducing numerical data
- This section need a re-writing in order to avoid repetition of similar words or verbs
- In general, the length is great, but feel the results could be expressed differently
Introduction:
- Line 30: Probably weeds cannot be considered as "pests"
- Many statements require a proper citation to validate them
- Line 34-36: This sentence is not correct, or it's confusing. Same for Line 41-43... I would recommend to carefully check the writing of the whole section
- In general, is very short, very few state-of-the art, not very well referenced, need a revision of the use of English, confusing in many phrases
Materials and methods:
- Definitely require more details and references
- Some approaches, as the fungi isolation or the extraction, are questionable. Not very confident
- Extractive compounds (EtOAc, DMSO) were tested? Later you explain how to test this, but in several parts of the text this is not clear, and they can also have antimicrobial effect themselves. Moreover, which amounts were extracted, which mixes... Very incomplete
- Identification methods... not even described... All this section really requires a deeper work
- Which are the pathogen strain codes? Type strains? Very short info about how to perform antimicrobial assays. Details are critical to detect their reliability
- Weed tests are ok, but why those species, why not others? Did you test in beneficial weeds or other crops to avoid indirect damage?
- Quantification indexes indicated here are not very accurate methods to evaluate..., specially to give so precise numbers in results
Results:
- 3.2, you don't have your own NMR structure output? This Figure 1 is your own? Very similar to equisetin, but how close is it? Derivates? Identity? Not enough
- Don't repeat methodology in this section
- 3.3, here you don't compare fraction result's with the pure extract results, so it's difficult to see how much of the extract result is due to that fraction. Could be interesting to use directly the raw extract in some cases
- Statistics are not present in many tables and data
- When compared to boscalid effect, equitesin require notably more amount to cause EC50, so take this into the account, not as good as the positive control (by far). Some statements you mention in discussion and conclusions could be not right
Discussion:
- Non-existent, this is very important. You can merge in results, but not avoid it
Conclusions:
- Many phrases are introduction, not conclusions (only at the end and it's not a conclusion)
- Rebuild this section
Comments on the Quality of English LanguageRequire to improve expressions, avoid reiterations (both in concepts and use of same/similar root words), etc.
Round 2
Reviewer 2 Report
Comments and Suggestions for Authors
I really appreciate the changes incorporated, but don't feel sure about many other that remain untouched since last version. Editors know my concerning and they will decide
